# The Study of Variation of Metabolites by Sleep Deficiency, and Intervention Possibility of Aerobic Exercise

**DOI:** 10.3390/ijerph19052774

**Published:** 2022-02-27

**Authors:** Jong-Suk Park, Young-Jun Kim, Wan Heo, Sangho Kim

**Affiliations:** 1School of Global Sport Studies, Korea University, Sejong-si 30019, Korea; model200@korea.ac.kr; 2Department of Food and Biotechnology, Korea University, Sejong-si 30019, Korea; yk46@korea.ac.kr; 3Department of Food Science and Engineering, Seowon University, Cheongju-si 28674, Korea; 01062033526@korea.ac.kr

**Keywords:** sleep deficiency, aerobic exercise, sleep-related factor, metabolites, metabolomics

## Abstract

The purpose of this study is to determine the difference in sleep-related factors and metabolites between normal sleep (NS) and sleep deficiency (SD) and to analyze the variations in metabolites according to the intensity of aerobic exercise under SD conditions. This study was conducted on 32 healthy male university students. Participants experienced both NS (8 h of sleep per night for 3 consecutive days) and SD (4 h of sleep per night for 3 consecutive days). After the SD period, the participants underwent treatment for 30 min by the assigned group [sleep supplement after SD (SSD), low-intensity aerobic exercise after SD (LES), moderate-intensity aerobic exercise after SD (MES), high-intensity aerobic exercise after SD (HES)]. For analysis, sleep-related factors were measured, and metabolites were analyzed by untargeted metabolite analysis using gas chromatography-time-of-flight mass spectrometry. As a result, SD showed that total sleep time (TST), duration of rapid eye movement (REM), duration of light sleep, and duration of deep sleep were significantly decreased compared to NS, whereas the Pittsburgh sleep quality index (PSQI), Epworth sleepiness scale (ESS), and visual analogue scale (VAS) were significantly increased compared to NS. The difference in metabolites between NS and SD showed that there were significant changes in the seven metabolites. There were 18 metabolites that changed according to the treatment groups in SD conditions. In summary, SD can exacerbate sleep quality, induce daytime sleepiness, increase fatigue, and increase metabolites that cause insulin resistance. Aerobic exercise under SD conditions can reduce metabolites that induce insulin resistance and increase the metabolites that help relieve depression caused by SD. However, HES has a negative effect, which increases fatigue, whereas LES has no negative effect. Thus, this study suggests that LES is the most appropriate exercise method under SD conditions.

## 1. Introduction

Sleep is critical to routine functioning and a high quality of life. Sleep modulates physiological functions, is significant for normal metabolic health, and is necessary for optimal health conditions. Sleep provides rest to the central nervous system and helps restore homeostasis [1]. In addition, sleep aids recovery from mental and physical fatigue from daytime activity and energy replenishment [2]. Sleep significantly contributes to memory function and neuronal plasticity [3]. Although sleep is critical, sleep duration has decreased in modern society. Given the excessive competition, individuals are exposed to sleep deficiency (SD) because of excessive stress, fatigue, and frequent voluntary or forced overworking. Furthermore, individuals often experience disturbances in their usual sleep patterns because of occupational characteristics, life conditions, aging, and diseases [4]. In humans, SD has been associated with several adverse health consequences. Previous studies have reported that SD is associated with an increased risk of obesity, diabetes, hypertension, metabolic syndrome, and cardiovascular disease [4,5,6,7,8,9]. Several epidemiologic studies have shown that a continued short sleeping time strongly correlates with increased risks of mortality and morbidity or diseases [10].

Metabolomics refers to the study of a comprehensive set of metabolites, that are context-dependent and vary depending on physiology and the pathological status of the cells, tissues, and organisms [11]. Metabolomics is a relatively new analytical method within the field of sleep medicine and exercise biochemistry. With the greater ease-of-use and availability of mass-spectrometry equipment, metabolomics has improved access to powerful data acquisition and advanced data processing techniques to detect large numbers of analytes simultaneously in biological samples [12]. Recently, the application of metabolomics in sleep studies has increased due to the advantages of metabolomics. Emerging advancements in metabolomics allow us to understand the complex pathophysiological mechanisms of the health effects of SD. To understand the mechanisms underlying the health effects of SD, total sleep deprivation (TSD), and obstructive sleep apnea, many recent epidemiologic studies have leveraged metabolomics [13,14,15]. Previous studies that used metabolomics have shown that acute TSD in humans can cause extensive disturbance in the plasma metabolic profile [13]. Moreover, a targeted metabolomics study of fasting plasma samples suggested that even short-term TSD can cause adverse metabolic effects, such as insulin resistance, in humans [16]. Chronic TSD has been shown to cause a significant decrease in cognitive function and extensive metabolic disturbance in mouse serum [17]. Several studies have examined changes in the metabolites of individuals experiencing SD. SD resulted in various changes in circulating metabolites, including increased levels of certain lipids and amino acids, and decreased carbohydrates [18]. In addition, SD is associated with pronounced shifts in the lipid metabolites [19,20]. To date, however, most studies have focused on the effects of TSD, and studies on the effects of SD are fewer than those on TSD. In real-life conditions, people experience SD much more than TSD. Thus, it is important to examine the impact of SD in real life.

Regular exercise is an interventional therapy used for the prevention and treatment of diseases such as obesity, metabolic syndrome, and cardiovascular disease in adults. The present guidelines for physical exercise to gain these positive effects recommend regular physical training (low-to-moderate intensity, 30–60 min/session, and 2–7 sessions per week) [21]. Given the positive effects of regular exercise on the body, exercise is recommended for health and leisure to not only those with SD but also those without. Recent studies have reported that exercise is associated with positive outcomes in health problems caused by SD. Regular exercise has been reported as an interesting non-pharmacologic treatment for sleep disorders, and regular moderate exercise can prevent 24 h SD-induced impairment of long-term memory [22]. In addition, appropriate physical activity in individuals with SD has positive effects on obesity risk [4], synaptic plasticity [23], and insomnia therapy [24]. 

Although the best solution for SD is to allow adequate sleep, various social and environmental factors cause sleep disturbance. Therefore, to resolve and prevent SD-related health issues, alternative interventions in the absence of adequate sleep are required. Exercise could be the intervention method. However, studies comprehensively investigating the efficiency and safety of aerobic exercise as an intervention for health problems caused by SD are lacking. In addition, the theoretical basis for the beneficial effects of exercise on SD or SD-induced stress is unclear. Few metabolomic studies on the association between SD and exercise exist.

Therefore, this study was conducted to identify two purposes. The first purpose was to identify the difference in sleep-related factors (sleep quantity and quality) and metabolite changes between NS (8 h of sleep per night for 3 consecutive days) and SD (4 h of sleep per night for 3 consecutive days). The second purpose was to analyze changes in the metabolites according to the difference in aerobic exercise intensity under SD conditions. Through this, we tried to confirm the efficiency and safety of aerobic exercise under SD conditions.

## 2. Materials and Methods

### 2.1. Participants

The study protocols were approved by the Institutional Review Board of Korea University (1040548-KU-IRB-16-286-A-1) before the experiment, and the experiment was conducted under the guidelines of the Institutional Review Board of Korea University in accordance with the guidelines of the Declaration Helsinki (1964). In addition, this study has been registered with the Clinical Research Information Service (CRIS), which is a public trials registry of the Republic of Korea (CRIS-KCT0006684).

All participants fully understood the experimental procedure, and they provided written informed consent before participating in the study. For this study, participants with a history of any of the following were excluded: (1) did not have a regular sleep–wake schedule; (2) had self-reported habitual sleep of less than 7 h during a recent week; (3) were smokers; (4) had circadian or sleep disorder; (5) suffered from drug or alcohol abuse; (6) had a muscular-skeletal or neurological disease; (7) had metabolic, chronic inflammatory, or cardiovascular disease. 

The sample size for this study was calculated using G*Power software (version 3.1.9.2). Applying repeated measured ANOVA within/between interaction, with the power (1-β) set at 0.95, the significance level (α) set at 0.05, and the effect size set at 0.4. As a result of applying this method, the number of proper participants was calculated at 28. However, considering a dropout rate of ten percent during the experiment, a total of 32 healthy males who were university students with normal body mass index (BMI) were recruited. Participants were randomly divided into four groups (*n* = 8 for each group): sleep supplement after SD (SSD), low-intensity aerobic exercise after SD (LES), moderate-intensity aerobic exercise after SD (MES), and high-intensity aerobic exercise after SD (HES). 

Table 1 summarizes the baseline characteristics of all variables according to experimental groups. Statistical analysis was performed to ensure homogeneity among experimental groups, and there were no significant differences among experimental groups in all factors (*p* > 0.05). Therefore, homogeneity of characteristics among the participants was secured.

### 2.2. Experimental Protocol

To determine the participants’ exercise intensity, participants were measured on maximal oxygen uptake by a graded exercise test using a treadmill one week before the normal sleep (NS) period. After measuring maximal oxygen uptake, all participants underwent two sleep conditions. Referring to previous studies [25,26], the first sleep condition was NS of 8 h (23:00 to 07:00) for 3 consecutive nights, whereas the second sleep condition was SD of 4 h (03:00 to 07:00) for 3 consecutive nights. During the SD period, participants spent time reading a book, watching television, or playing internet games before sleeping. After a 3-day SD period, the participants performed 30 minutes of treatment according to their assigned group. For blood analysis, the participants’ blood samples were collected three times—after NS, after SD, and after group treatment (AT) (see Figure 1). 

During the experiment periods, napping and excessive physical activity during the daytime were not allowed, and this was monitored by a wearable activity tracker (Fitbit Charge 2, Fitbit Inc., San Francisco, CA, USA). Additionally, the participants were provided dinner of the same calories (900 kcal) at 7 pm and limited the intake of any food or drinks other than water after dinner during the experimental period. In addition, the researcher supervised using continuous telephone calls and mobile messages to identify whether they followed experiment control. The participants were prohibited from ingesting any alcohol, medication, or caffeine beverages as well as from doing excessive exercise during experiment periods. To measure temperature and relative humidity in the room, a digital thermometer and a hygrometer (Acuba cs-201, Chosun, Guangdong, China) were used. During the experiment period, the participant’s room temperature and relative humidity were kept at 24~25 °C and 40~50%. Additionally, the room was turned off and the outside light was blocked during sleep.

### 2.3. Treadmill Test

To establish the exercise intensity of the participants assigned to the exercise groups, the participant’s maximum oxygen uptake was measured one week before the NS period. After overnight fasting before the experiment, all experiments were conducted at the same time (09:00 to 11:00). The participants performed the maximal exercise using a treadmill (Cosmed T150, h/p Cosmos, Nussdorf-Traunstein, Germany) and autonomous respiratory gas analyzer (TrueOne 2400, ParvoMedics, Inc., Salt Lake City, UT, USA), and exercise protocol used was the Bruce protocol, which is a suitable exercise protocol for healthy adults. During the exercise, the researchers observed heart rate (HR) and results of the gas analysis [oxygen consumption (VO_2_), ventilation (V), respiration exchange ratio (RER), and respiratory rate (RR)], and recorded them. In this study, objective criteria for VO_2max_ determination were used: (a) HR > 90% of HR_max_ (220-age); (b) RER > 1.15 during the test; (c) a plateau of the oxygen uptake curve; (d) demand by participant due to fatigue. Meanwhile, all tests were performed at the same room temperature (23~24 °C) and relative humidity (50~55%).

### 2.4. Measurement of Sleep-Related Factors

The sleep-related factors of participants were measured twice (after NS and after SD). To measure the quantity of sleep and amount of daily activity, participants were provided a wearable activity tracker (Fitbit Charge 2, Fitbit Inc., San Francisco, CA, USA), and they wore it on the wrist of the non-dominant hand during the experimental period. Fitbit Charge 2 is a wristband that tracks sleep and all-day activity and has a built-in triaxial accelerometer, heart rate tracker, altimeter, and vibration motor. Fitbit devices demonstrate 98% sensitivity to identifying sleep episodes compared to polysomnography, with inter-device reliability >96% [27]. Through the Fitbit device, total sleep time (TST), time of each sleep stage, amount of daily activity, and calorie consumption were measured during the experimental period. The mean of the values measured for three days was used for the analysis. 

To measure sleep quality, daytime sleepiness, and fatigue during the experiment period, this study used the Pittsburgh Sleep Quality Index (PSQI), Epworth sleepiness scale (ESS), and visual analog scale (VAS). The PSQI consists of a nineteen-item self-rated questionnaire for evaluating subjective sleep quality. The total PSQI score is summed of seven component scores, which have a range of 0–21, and high scores indicate worse sleep quality. The PSQI has a specificity of 86.5% and a sensitivity of 89.6% for detecting cases with a sleep disorder, using a cut-off score of 5 [28]. The ESS is a widely used scale that evaluates the degree of sleepiness. The ESS score represents the sum of individual items and ranges from 0–24. Values > 10 are considered to indicate significant sleepiness [28]. The VAS is a simple method to measure the degree of subjective fatigue [29]. Typically, a VAS consists of a 10 cm line. The left end of the line is labeled with the anchor words “I do not feel tired” and the right with “I do feel extremely tired, exhausted”. The VAS is calculated by measuring the distance of the mark from the left of the line in centimeters. The values have a range between 0 and 10 and higher values mean higher levels of fatigue intensity. In this study, the participants were asked to mark a dot on the line that corresponded with their fatigue.

### 2.5. Treatment of Groups

Treatment by groups was performed at 9 am on the 4th day after the subject experienced SD for 3 days. Based on the VO_2max_ of participants after the SD period, participants performed low-intensity (40% of VO_2max_), moderate-intensity (60% of VO_2max_), high-intensity (80% of VO_2max_) aerobic exercise for 30 min assigned to each group. Using the Bruce protocol, at the time of reaching the targeted exercise intensity allocated to the participants during the exercise, the exercise was performed by maintaining the speed and grade of the treadmill. If the exercise intensity during exercise was higher or lower than the exercise intensity assigned to the participant, the researcher adjusted the speed without adjusting the grade to maintain the participant’s exercise intensity. We also measured exercise-related factors (VO_2_, HR, RER, running speed, calorie consumption, and running distance) to ensure that participants exercised correctly for each group they were assigned to. All tests were performed at the same room temperature (23~24 °C) and relative humidity (50~55%).

On the other hand, The SSD group was treated with sleep supplementation after the SD period. Referring to previous studies [30,31], they slept on a bed in the laboratory to supplement their sleep for 30 min after resting. At this time, the laboratory turned off the indoor lights and blocked the external noise and outdoor lights by using a black curtain. Additionally, room temperature and humidity were set at 24~25 °C and 40~50%.

### 2.6. Metabolomics

#### 2.6.1. Blood Sample Collection

For the metabolite analysis, a certified medical technologist collected 10 mL of blood from each participant’s median cubital vein three times (rest after NS, rest after SD, and after treatment of each group), using a vacutainer serum tube (BD367820, Becton Dickinson, NJ, USA). Blood samples were clotted at room temperature for 20 min and centrifuged at 3000 rpm for 15 min at room temperature using a centrifuge (Union 32R, Hanil Co., Incheon, Korea) to separate the serum. The collected serum was stored immediately at −80 °C until analysis.

#### 2.6.2. Chemicals and Materials

Methanol was purchased from Fisher Scientific (Pittsburgh, PA, USA). Pyridine, methoxyamine-hydrochloride, *N*-methyl-*N*-(trimethylsilyl) trifluoroacetamide (MSTFA), and standard compounds were obtained from Sigma Chemical Co. (St. Louis, MO, USA).

#### 2.6.3. Metabolite Extraction

For serum extraction, 1 mL of methanol containing an internal standard (2-chlorophenylalanine 1 mg/1 mL) was added to 200 μL of serum, and the mixture was sonicated with shaking for 20 min. The samples were centrifuged (15,000× *g*) for 10 min at 4 °C. The partially purified supernatant was filtered using a 0.2-μm polytetrafluoroethylene filter and concentrated using a speed vacuum concentrator (Modulspin 31, Biotron, Korea).

#### 2.6.4. Gas Chromatography-Time-of-Flight-Mass Spectrometry (GC-TOF-MS) Analysis

This study was performed untargeted metabolomics to simultaneously measure as many metabolites as possible from serum using GC-TOF-MS analysis. GC-TOF-MS analysis was performed using a gas chromatograph system (Agilent 7890, Agilent Technologies, Palo Alto, CA, USA) coupled with an auto-sampler (Agilent 7693, Agilent Technologies, Palo Alto, CA, USA) and equipped with Pegasus® HT-TOF-MS (LECO, St. Joseph, MI, USA) system. An Rtx-5MS column (30 m × 0.25 mm × 0.25 μm particle size, Restek Corp., Bellefonte, PA, USA) was used with a continuous flow of 1.5 mL/min of helium as carrier gas. One μL of the derivatized samples was placed in an auto-sampler and the sample was injected with splitless mode. The ion source, front inlet and transfer line temperatures were set at 230 °C, 250 °C, and 240 °C, respectively. The oven temperature was maintained at 75 °C for 2 min, then increased to 300 °C at 15 °C/min, and then maintained at the final temperature for 3 min. Electron ionization was performed at 70 eV, and mass data was collected by full scanning over a mass-to-charge ratio (*m*/*z*) range of 50–600. The analytical samples were analyzed in blocks of 10 runs followed by an intermittent quality control sample (QC, with 10 μL pooled blends from all samples) to decrease systematic error. All MS analyses were randomized in each block. These methods were referred to in previously described analytical methods [32].

### 2.7. Data Processing and Statistical Analysis

The statistical analysis of sleep-related factors was performed using SPSS software (version 24.0, IBM Corp., Armonk, NY, USA). To ensure that all data had a normal distribution, the Shapiro–Wilk test was used. All data of this study identified a normal distribution, data is expressed as a mean ± standard deviation. Significant differences in sleep-related factors between NS and SD were compared using a paired *t*-test, and significant differences in the characteristics and quantity of exercise according to treatment groups after SD were analyzed using one-way analysis of variance (ANOVA) following Tukey’s test.

To analyze metabolite data processing and multivariate statistical analysis, GC-TOF-MS raw data were preprocessed, acquired, and converted into the NetCDF format (*.cdf) using the LECO Chroma TOFTM software (version 4.44, LECO Corp., St. Joseph, MI, USA). After the conversion, peak detection, retention time correction, and alignment were processed using the MetAlign software package. The data were exported to an Excel file. To identify endogenous metabolites, the multivariate statistical analysis was performed using SIMCA-P+ (version 12.0, Umetrics; Umea, Sweden). Principal component analysis (PCA) and orthogonal projections to latent structures discriminant analysis (OPLS-DA) were performed to compare metabolite differences between NS and SD or among treatment groups. The significantly different metabolites were selected based on variable importance in the projection (VIP) value >0.7 and *p* values <0.05 in OPLS-DA to identify significantly different metabolites. The selected metabolites were tentatively identified based on various information such as retention time, mass spectra, and mass fragment pattern obtained from GC-TOF-MS as well as matching with standard compound analyzed under the same condition, and databases including the National Institutes of Standards and Technology (NIST) library, the Human Metabolome Database. Additionally, to identify significant differences between NS and SD as well as between SD and each treatment group, the variables were analyzed by an independent t-test and one-way analysis of variance (ANOVA) following Tukey’s test. The statistical significance levels (α) of all the analyses were set to 0.05.

## 3. Results

### 3.1. The Differences in Sleep-Related Factors between NS and SD

The differences in sleep-related factors between NS and SD are shown in Table 2. The TST, duration of REM, duration of light sleep, and duration of deep sleep in SD were significantly decreased compared to NS (*p* < 0.001). However, PSQI, ESS, and VAS in SD had a significant increase compared to NS (*p* < 0.001), while the amount of daily activity and the amount of calorie consumption were not significantly different between NS and SD (*p* > 0.05).

### 3.2. The Differences in Serum Metabolites between NS and SD

To obtain the statistically significant serum metabolite between NS and SD, multivariate statistical analysis was conducted using PCA and OPLS-DA. For quality control (QC), eight random mixtures of all the samples were injected after every six or seven samples as a means to select outlying samples. A PCA was carried out to show the correlation between the observations. As shown in Figure 2A, NS and SD were unclearly divided into two clusters on the PCA score plot. Thus, OPLS-DA, which is supervised methods, was further constructed to intensify the discriminant between NS and SD. As shown in Figure 2B, OPLS-DA score plot showed a significantly clear separation between NS and SD (*p* = 0.002). The R2Y (cum), indicating fitness, was 0.871, and Q2 (cum), indicating prediction accuracy, was 0.393 in this OPLS-DA model. 

From this calculated OPLS-DA, a total of 28 metabolites including 10 amino acids, 8 sugars and sugar alcohols, 2 fatty acids, and 8 non-identifications were significantly affected by SD. The 28 metabolites were searched through the NIST library and HMDB based on accurate mass values, mass fragment pattern, and structure information. Their information is summarized in Table 3. As shown in Table 3, the relative levels of leucine, isoleucine, proline, serine, ornithine, glucose, saccharide 4, palmitic acid, and stearic acid were increased by SD. In contrast, the fructose, galactose, saccharide 2, and saccharide 3 levels were decreased by the SD. The 7 metabolites were identified and those had statistically significant differences having VIP > 0.7 and *p* < 0.05. As shown in Table 3, the relative levels of leucine (1.115-fold, *p* < 0.001), isoleucine (1.105-fold, *p* < *0*.01), glucose (1.129-fold, *p* < 0.05), saccharide 4 (1.317-fold, *p* < 0.001) and stearic acid (1.120-fold, *p* < 0.05) were significantly increased by SD. In contrast, galactose (0.734-fold) and saccharide 3 (0.779-fold) levels were significantly decreased by SD (*p* < 0.05).

### 3.3. The Differences in Exercise-Related Factors among Exercise Groups

The average of the VO_2_, RER, HR, running speed, running distance, and calorie consumption measured during exercise in the treatment groups are shown in Table 4. All factors had significant differences among the exercise groups (*p* < 0.001). In all measurement variables, LES was significantly lower and HES was significantly higher. Thus, these results showed that exercise was well performed to the assigned exercise intensity of each exercise group.

### 3.4. The Differences in Serum Metabolites between SD and AT

To obtain the statistically significant serum metabolite between SD and AT, multivariate statistical analysis was conducted using PCA and OPLS-DA. For QC, eight random mixtures of all the samples were injected after every six or seven samples as a means to select outlying samples. A PCA was conducted to show the correlation between the observations. As shown in Figure 3A, SD and treatment groups were unclearly divided into two clusters on the PCA score plot. Thus, OPLS-DA was further constructed to intensify the discriminant between SD and AT. As shown in Figure 3B, OPLS-DA score plot showed a significantly clear separation between SD and AT (*p* < 0.001). The R2 (cum), indicating fitness, was 0.592 and Q2 (cum), indicating prediction accuracy, was 0.609 in this OPLS-DA model.

From this calculated OPLS-DA, a total of 35 metabolites including 3 organic acids, 10 amino acids, 9 sugars and sugar alcohols, 5 fatty acids and lipids, and 8 non-identifications were significantly affected by treatment groups. The 35 metabolites were searched through the NIST library and HMDB based on accurate mass values, mass fragment pattern, and structure information. Their information is summarized in Table 5. As shown in Table 5, 18 metabolites were identified and those had statistically significant differences having VIP > 0.7 and *p* < 0.05. The 18 metabolites were 3 organic acids (lactic acid, oxalic acid, and phosphoric acid), 5 amino acids (valine, glycine, lysine, tyrosine, and tryptophan), 7 sugars and sugar alcohols (fructose, galactose, glucose, glucose, saccharide 2, saccharide 4, and myo-inositol) as well as 3 fatty acids and lipids (palmitic acid, oleamide, and cholesterol). 

In the SSD, five metabolites (oxalic acid, valine, lysine, fructose, and saccharide 2) were significantly higher compared with SD (*p* < 0.05). However, one metabolite (saccharide 4) was significantly lower compared with SD (*p* < 0.05). In the LES, eight metabolites (lysine, tyrosine, tryptophan, fructose, saccharide 2, myo-Inosistol, palmitic acid, and oleomide) were significantly higher compared with SD (*p* < 0.05). However, three metabolites (phosphoric acid, glucose, and saccharide 4) were significantly lower compared with SD (*p* < 0.05). In the MES, five metabolites (glycine, lysine, saccharide 2, myo-inositol, and palmitic acid) were significantly higher compared with SD (*p* < 0.05). However, one metabolite (phosphoric acid) was significantly lower compared with SD (*p* < 0.05). In the HES, nine metabolites (lactic acid, oxalic acid, glycine, lysine, fructose, galactose, saccharide 2, myo-inositol, and cholesterol) were significantly higher compared with SD (*p* < 0.05). However, three metabolites (phosphoric acid, glucose, and saccharide 4) were significantly lower compared with SD (*p* < 0.05).

## 4. Discussion

In humans, both total sleep duration and sleep quality are important factors to maintain health. In this study, the participants experienced both NS and SD, each for 3 days. To evaluate the status of sleep, this study measured both sleep time and quality of sleep using a wearable activity tracker and a self-reported questionnaire with proven reliability and validity. PSQI is generally used as a tool to assess sleep quality, and ESS is a measure indicating daytime sleepiness. Furthermore, VAS is a tool that can evaluate the degree of fatigue [28,29,33]. According to our findings, SD resulted in a significant decrease in TST. In addition, with regard to the observed changes in sleep stages, duration of REM sleep, duration of light sleep and duration of deep sleep were significantly decreased. In this study, along with the changes in sleep duration and architecture, PSQI, ESS, and VAS were also significantly increased after SD. These results agree with those of previous studies [34,35,36,37,38,39,40] and suggest that quantitative and qualitative SD decrease sleep quality, induce daytime sleepiness, and increase fatigue. In addition, it means that the SD condition, which is essential to confirm the metabolite change in SD conditions, is well induced. In contrast, the amount of energy intake and physical activity did not differ significantly between NS and SD during the study period. These results indicate that energy intake and physical activity, which were confounding factors that could affect the outcome of the experiment, were well controlled.

The branched chained amino acids (BCAAs), including leucine, isoleucine, and valine, are the only amino acids metabolized in the muscle and function as hormonal signaling indicators in addition to serving as nutrients [41]. During stressful conditions or exercise, BCAAs act as an important substrate and serve as a precursor to the synthesis of other amino acids and proteins [42]. In this study, the increase in leucine and isoleucine levels was greater after SD than after NS. This is consistent with the results reported by previous study [18]. They showed that sleep restriction caused an elevation in multiple plasma amino acids and associated metabolites. In addition, previous studies have suggested that SD has catabolic effects on protein metabolism and lean body mass [43,44]. Sleep loss may necessitate additional energy intake for glucose-dependent tissues via increased protein breakdown and muscle loss [44]. In addition, in a stressful situation, the adrenal cortex secretes cortisol, which breaks down proteins and fats to supply the body with the energy needed to cope with stress by increasing blood glucose levels. In addition, excessive cortisol secretion because of persistent stress causes complications such as excessive protein degradation, nitrogen balance inhibition, and immune loss [45]. Therefore, SD is believed to increase crease protein metabolism through complex pathways, such as changes in energy substrate response to energy demands due to prolonged wakefulness conditions and cortisol increase due to stressful conditions. 

With regard to the association between SD and insulin resistance, several studies have demonstrated that SD has detrimental effects on insulin sensitivity and glucose tolerance, and causes increased insulin resistance, which is an indicator of prediabetes [46,47,48]. Both sleep duration and quality may affect glucose metabolism [47,49]. In physiological conditions, glucose utilization is the highest during wakefulness and the lowest during REM sleep. Previous studies showed that sleep restricted to only 4 h for two or more nights decreased glucose tolerance by 40% and decreased acute insulin response to glucose by 30% in healthy individuals [7,50]. In this study, we observed that blood glucose levels were higher after SD than after NS. This result is consistent with results from the aforementioned previous studies. Previous studies asserted that sleep loss-induced changes in the somatotropic, adrenocortical, and sympathetic nervous systems may contribute to the development of glucose tolerance [50,51], and the development of insulin resistance is caused by oxidative stress, proinflammatory, and endoplasmic reticulum stress pathways during SD [52,53]. Another plausible explanation is exposure to light. Light exposure stimulates glucocorticoid secretion and increases plasma glucose [54,55]. To limit the participants’ sleep in this study, participants were exposed to light while they were spending time through reading, playing computer games, or watching movies. In addition, the participants had no additional food intake until sleep was induced after dinner during the experiment period, and they had to fast until blood collection. Nevertheless, fasting glucose after SD is believed to have increased because of abnormalities in insulin function such as insulin resistance. 

Stearic acid is a saturated fatty acid, and it has been associated with the incidence of type 2 diabetes. A recent study reported that stearic acid was positively correlated with the C-peptide level, which is an efficient marker for insulin resistance [56]. In addition, long-chain saturated fatty acids such as stearic acid are known to induce lipotoxicity, which causes cellular dysfunction [57]. Lipotoxicity causes beta-cell damage and plays a crucial role in the development of type 2 diabetes [58,59]. In this study, the stearic acid level after SD was significantly higher than that after NS. Although the mechanism underlying stearic acid level increase by SD remains unknown, our results suggest that the increase in stearic acid levels after SD is closely associated with insulin resistance. This result means that persistent SD could cause type 2 diabetes due to insulin resistance and lipotoxicity. 

We used metabolomics to confirm metabolic changes according to the treatment groups after SD. We observed that glucose levels in LES and HES were significantly decreased compared to SD. Muscle contraction and insulin are the primary physiological stimulators of glucose transport in skeletal muscle [60]. In addition, a previous study suggested that exercise increases insulin sensitivity [61]. Therefore, similar to the effects of exercise in normal conditions, the results of this study suggest that aerobic exercise in SD conditions also decreases blood glucose concentration through an increase in blood flow and the ability of glucose transporter-4 to transport glucose across the cell membrane. Therefore, aerobic exercise is important for glucose metabolism in skeletal muscle under SD conditions. In addition, glycine is a non-essential amino acid, and is known to be low in patients with diabetes. A previous study showed that higher glycine level is associated with decreased risk of type 2 diabetes [62]. In this study, the changes in glycine levels in the exercise groups (MES and HES) after SD significantly increased compared to rest after SD. Therefore, this result indicates that aerobic exercise after SD can reduce the risk of type 2 diabetes caused by SD.

Tyrosine is a dopamine precursor, which is synthesized only in the brain. Dopamine, a type of catecholamine, mainly affects behavior, attention, and learning [63], and is known to act as a major pathologic mechanism that causes multiple brain diseases, such as attention deficit hyperactivity disorder (ADHD), Parkinson’s disease, and schizophrenia [64]. Dopamine is strongly linked to the regulation of sleep and wakefulness, indicating an inverse association between dopamine biosynthesis and the quantity of sleep [65]. In addition, tryptophan is an amino acid well known to be a precursor of serotonin and melatonin [66]. Tryptophan is essential for the formation of serotonin and melatonin via the indoleamine pathway [13] and it has been used to treat depressive disorders [67]. SD is frequently associated with depression [68,69], depression is known to be associated with functional impairment of serotonergic neurotransmitter systems, such as tryptophan hydroxylase and serotonin receptors [70]. Another important factor in depression is the change in dopamine levels and dopamine receptors [71]. That is, reduced serotonin and dopamine levels are closely associated with depression [72]. Although no significant differences were observed in this study, SD reduces the release of serotonin and dopamine [73], leading to depression. That is, reduced serotonin and dopamine levels are closely associated with depression [72]. Interestingly, we showed that tyrosine and tryptophan levels only in the LES were significantly higher than those in SD. Several studies have reported similar results that exercise increases serotonin and dopamine levels [74,75]. The increased levels of tyrosine and tryptophan in LES may contribute to the antidepressive effect, directly or indirectly via dopamine synthesis and serotonin synthesis. Thus, this result suggests that LES could alleviate daytime sleepiness by increasing tyrosine levels and prevent depression caused by SD through the metabolic mechanisms of increased tyrosine and tryptophan levels.

Myo-inositol, a stereoisomer of inositol, plays a key role in critical intracellular signaling pathways. Recently, a previous study reported that myo-inositol is a potential biomarker for depression [76]. Patients with depression are associated with reduced levels of myo-inositol [77], and SD is closely associated with depression [69]. In this study, the myo-inositol levels significantly increased in all exercise groups. Thus, aerobic exercise can help alleviate depression caused by SD by increasing myo-inositol level through aerobic exercise. 

The levels of some fatty acids such as palmitic acid and oleamide significantly increased in the exercise groups after SD. This could likely be attributed to serotonin activity. Exercise increases serotonergic activity, serotonin synthesis during exercise is mainly due to increased adrenergic-mediated stimulation of lipolysis, which increases plasma free fatty acids (FFAs) [73]. Therefore, increased FFAs levels may increase serotonin, which contributes to sleep and the prevention of depression [78]. 

Lactic acid is an organic acid that contributes to several biochemical processes, and it is produced in the muscles during intense activity. Lactic acid is metabolized through anaerobic metabolism of glucose during exercise. It is produced when the ATP concentration in muscle cells exceeds that resynthesized in mitochondria and acts as the main limiting factor during exercise [79]. In this study, the lactic acid level showed that HES was higher than at rest after SD. This result suggests that HES causes fatigue. 

### Limitations

There are several limitations to consider when interpreting our results. First, SD stimulation was applied for 3 days in this study. Thus, it is difficult to apply the results to chronic SD or insomnia patients. Second, because the aerobic exercise was only performed once in this study, it will be different from the effects of long-term training and anaerobic exercise. Lastly, all participants of the exercise groups exercised equally for 30 min in this study. Thus, there may be a difference from the effect of long-term exercise.

## 5. Conclusions

In summary, SD can attenuate the quality of sleep, induce daytime sleepiness, and increase fatigue. In addition, SD can increase metabolites that cause insulin resistance. However, in SD conditions, aerobic exercise can reduce metabolites that cause insulin resistance, and increase the metabolites that help relieve depression. However, high-intensity aerobic exercise in SD conditions has a negative effect, which increases fatigue, whereas low-intensity aerobic exercise has the most positive effect on metabolites. Therefore, these findings suggest that low-intensity aerobic exercise is the most appropriate exercise method in SD conditions. Based on the limitations of this study, the following studies should be conducted in future studies. First, future studies in patients with chronic SD and insomnia should be conducted. Second, it should be done to verify the effects of long-term training and anaerobic exercise. Lastly, a study should be conducted to confirm the difference when each group performs with the same amount of exercise.

## Figures and Tables

**Figure 1 ijerph-19-02774-f001:**
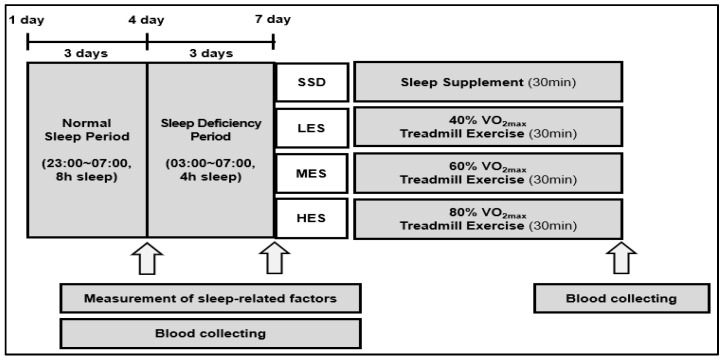
Experimental procedure.

**Figure 2 ijerph-19-02774-f002:**
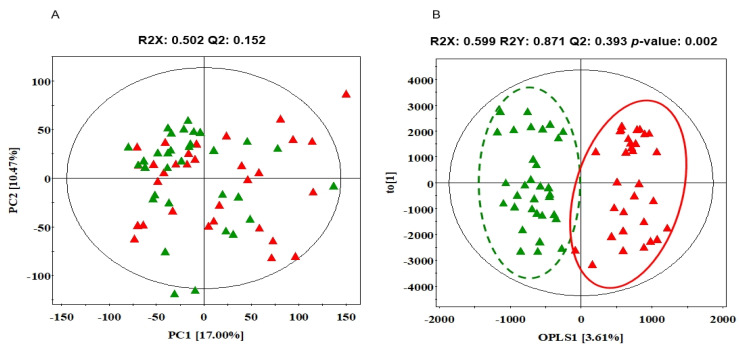
PCA and OPLS-DA score scatter plot based on GC-TOF-MS chromatographic data between NS and SD. PCA score scatter plots (**A**) and OPLS-DA score scatter plots (**B**) of serum extracts analyzed by GC-TOF-MS. Red filled triangles, NS; green filled triangles, SD. GC-TOFMS: gas chromatography-time-of-fight mass spectrometer, PCA: principal component analysis, OPLS-DA: orthogonal partial least squares discriminant analysis, NS: normal sleep, SD: sleep deficiency.

**Figure 3 ijerph-19-02774-f003:**
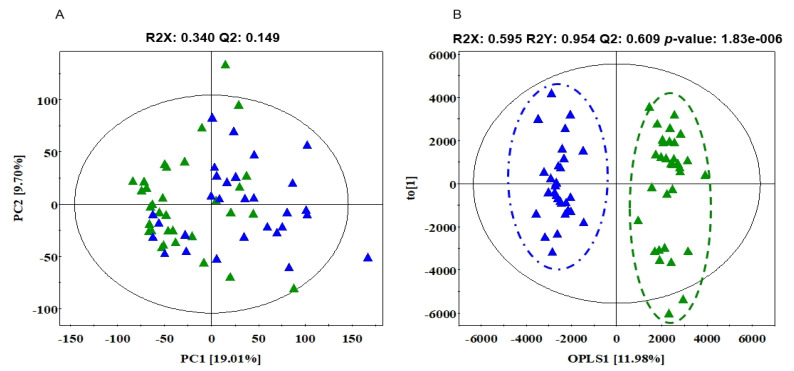
PCA and OPLS-DA score scatter plot based on GC-TOF-MS chromatographic data between SD and AT. PCA score scatter plots (**A**) and OPLS-DA score scatter plots (**B**) of serum extracts analyzed by GC-TOF-MS. Green filled triangles, SD; blue filled triangles, AT. GC-TOFMS: gas chromatography-time-of-fight mass spectrometer, PCA: principal component analysis, OPLS-DA: orthogonal partial least squares discriminant analysis, SD: sleep deficiency, AT: after treatment.

**Table 1 ijerph-19-02774-t001:** Characteristics of the participants according to groups and results of homogeneity.

Variables	SSD (*n* = 8)	LES (*n* = 8)	MES (*n* = 8)	HES (*n* = 8)	*F*	*p*
Age (years)	20.38 ± 0.52	21.88 ± 2.10	21.38 ± 2.00	21.50 ± 1.77	1.172	0.361
Height (cm)	176.88 ± 4.36	176.88 ± 3.27	174.63 ± 4.47	174.13 ± 5.19	0.887	0.460
Weight (kg)	75.23 ± 8.82	78.68 ± 11.78	70.38 ± 7.13	70.78 ± 5.74	1.657	0.199
BMI (kg/m^2^)	24.03 ± 2.14	25.06 ± 2.95	23.05 ± 1.63	23.36 ± 2.21	1.216	0.322
Muscle mass (kg)	35.26 ± 3.41	35.51 ± 3.22	33.38 ± 3.79	33.70 ± 2.65	0.863	0.472
Body fat percentage (%)	17.40 ± 3.50	19.90 ± 5.16	16.40 ± 2.23	16.09 ± 6.03	1.193	0.331
SBP (mmHg)	114.75 ± 6.25	114.00 ± 9.61	115.63 ± 9.01	116.50 ± 6.33	1.051	0.385
DBP (mmHg)	74.50 ± 5.86	71.63 ± 7.87	74.13 ± 5.72	71.38 ± 7.44	0.464	0.710
VO_2max_ (ml/kg/min)	53.33 ± 4.57	52.12 ± 6.77	51.79 ± 5.37	53.29 ± 7.40	0.134	0.939
PSQI (point)	4.13 ± 0.64	3.25 ± 1.28	3.25 ± 0.71	3.25 ± 1.48	1.285	0.299
ESS (point)	4.00 ± 0.93	3.13 ± 1.46	4.25 ± 1.03	3.63 ± 0.92	1.567	0.219

Data are expressed as mean ± standard deviation. A one-way ANOVA was used to analyze differences among treatment groups. SSD: sleep supplement after sleep deficiency, LES: low-intensity aerobic exercise after sleep deficiency, MES: moderate-intensity aerobic exercise after sleep deficiency, HES: high-intensity aerobic exercise after sleep deficiency, BMI: body mass index, SBP: systolic blood pressure, DBP: diastolic blood pressure, PSQI: Pittsburgh Sleep Quality Assessment, ESS: epworth sleepiness scale.

**Table 2 ijerph-19-02774-t002:** The differences in sleep-related factors between NS and SD.

Variables	NS	SD	t	*p*
TST (min)	400.90 ± 17.32	216.28 ± 16.56	54.710	<0.001
Duration of REM (min)	80.89 ± 17.51	37.16 ± 11.26	13.905	<0.001
Duration of light sleep (min)	248.33 ± 28.52	138.14 ± 18.11	23.941	<0.001
Duration of deep sleep (min)	71.68 ± 18.28	40.99 ± 11.58	10.449	<0.001
Steps during the day (steps)	10,637.06 ± 3549.80	10,961.25 ± 2644.61	−0.514	0.611
Calorie consumption (kcal)	2884.61 ± 455.95	2907.91 ± 343.72	−0.371	0.713
PSQI (points)	3.13 ± 1.39	6.16 ± 1.35	−11.333	<0.001
ESS (points)	4.34 ± 2.31	11.00 ± 4.56	−9.012	<0.001
VAS (points)	3.64 ± 1.50	7.32 ± 1.17	−13.118	<0.001

Data are expressed as mean ± standard deviation. Paired *t*-test was used to analyze difference between NS and SD. NS: normal sleep, SD: sleep deficiency, TST: actual total sleep time, REM: rapid eye movement, PSQI: Pittsburgh sleep quality index, ESS: Epworth sleepiness scale, VAS: visual analogue scale.

**Table 3 ijerph-19-02774-t003:** List of twenty-eight metabolites altered by SD.

No	Ret (min)	Metabolites	Unique Mass	Fragment Pattern (*m*/*z*)	ID	Fold Change (SD/NS)
Amino acids						
1	6.78	Valine	218	73, 144, 218, 147, 100, 74, 146, 219, 156	STD/MS	1.034
2	7.32	Leucine	158	73, 158, 147, 159, 75, 103, 117, 160	STD/MS	1.115 ***
3	7.54	Isoleucine	158	73, 158, 218, 147, 159, 100, 75, 74	STD/MS	1.105 **
4	7.59	Proline	142	142, 73, 143, 75, 147, 74, 70, 66, 144	STD/MS	1.187
5	7.67	Glycine	86	73, 174, 86, 147, 100, 175, 59, 248, 176	STD/MS	0.953
6	8.13	Serine	218	73, 204, 218, 100, 147, 75, 205, 74	STD/MS	1.082
7	8.40	Threonine	219	73, 57, 117, 101, 219, 147, 100, 129	STD/MS	1.042
8	9.58	*trans*-4-hydoxyl-L-proline	156	156, 73, 147, 157, 230, 75, 258, 14, 158	STD/MS	0.971
9	10.27	Ornithine	142	73, 70, 142, 74, 75, 102, 147, 144	STD/MS	1.288
10	10.41	Phenylalanine	218	73, 218, 192, 100, 147, 75, 74, 219, 193	STD/MS	1.041
Sugars and Sugar alcohols						
11	12.09	Saccharide 1	73	73, 147, 217, 191, 103, 129, 218, 75	MS	0.977
12	12.26	Fructose	103	73, 103, 217, 147, 307, 74, 133, 75	STD/MS	0.727
13	12.39	Galactose	204	73, 204, 191, 147, 205, 217, 129, 75, 103	STD/MS	0.734 *
14	12.48	Glucose	205	73, 147, 205, 160, 103, 217, 319, 74	STD/MS	1.046
15	12.60	Glucose	205	73, 147, 103, 205, 160, 129, 217, 157, 319	STD/MS	1.129 *
16	12.81	Saccharide 2	217	73, 217, 75, 147, 103, 129, 74, 117, 59	MS	0.877
17	12.94	Saccharide 3	204	73, 204, 147, 191, 217, 75, 205, 74, 129	MS	0.779 *
18	13.27	Saccharide 4	204	73, 204, 147, 205, 75, 217, 74, 129	MS	1.317 ***
Fatty acids						
19	13.22	Palmitic acid	313	117, 75, 73, 132, 129, 131, 118, 133, 313	STD/MS	1.097
20	14.40	Stearic acid	341	117, 73, 132, 129, 145, 131, 133, 118, 341	STD/MS	1.120 *
Non-identification						
21	4.14	N.I.1	151	171, 73, 78 64, 172, 151, 173, 186	-	0.709
22	6.03	N.I.2	86	73, 147, 133, 59, 86, 100, 72, 89	-	0.794 **
23	6.20	N.I.3	147	147, 73, 117, 75, 66, 191, 88, 148	-	1.086
24	7.64	N.I.4	107	107, 73, 77, 256, 55, 84, 140	-	0.901
25	7.94	N.I.5	184	184, 73, 134, 77, 86, 59, 100, 285	-	0.848
26	9.89	N.I.6	147	73, 147, 177, 292, 220, 103, 102, 130	-	0.247 *
27	10.16	N.I.7	227	73, 227, 147, 155, 154, 139, 59, 75	-	1.280 **
28	11.44	N.I.8	299	73, 299, 147, 357, 103, 101, 75, 129	-	0.730 **

Metabolites selected by VIP >0.7 and *p* value (<0.05) from OPLS-Da. * *p* < 0.05, ** *p* < 0.01 ***, *p* < 0.001 a significant difference between NS and SD. NS: normal sleep, SD: sleep deficiency, OPLS-DA: orthogonal partial least squares discriminant analysis, Ret: retention time, ID: identification, STD: commercial standard compound, MS: to comparison with the mass spectra, *m*/*z*: the selected ion for identification and mass-to-charge ratio.

**Table 4 ijerph-19-02774-t004:** The differences in VO_2_, HR, RER, running speed, running distance and calorie consumption according to exercise groups.

Variables	LES (a)	MES (b)	HES (c)	*F*	*p*	Post Hoc
VO_2_ (ml/kg/min)	21.25 ± 2.74	33.71 ± 2.96	43.31 ± 4.93	72.320	<0.001	a < b < c
HR (beat)	120.60 ± 9.36	145.41 ± 9.28	172.15 ± 2.83	87.783	<0.001	a < b < c
RER	0.82 ± 0.02	0.85 ± 0.02	0.89 ± 0.04	12.760	<0.001	a < b < c
Running speed (mph)	2.32 ± 0.32	3.08 ± 0.22	4.16 ± 0.65	35.651	<0.001	a < b < c
Calorie consumption (kcal)	250.25 ± 25.52	352.75 ± 56.53	455.88 ± 48.32	41.033	<0.001	a < b < c
Running distance (m)	2297.25 ± 250.89	2912.25 ± 142.36	3651.38 ± 457.08	37.766	<0.001	a < b < c

Data are expressed as mean ± standard deviation. A one-way ANOVA was used to analyze differences among treatment groupsLES, low-intensity aerobic exercise after sleep deficiency; MES, moderate-intensity aerobic exercise after sleep deficiency; HES, high-intensity aerobic exercise after sleep deficiency; HR, heart rate; RER, respiratory exchange ratio.

**Table 5 ijerph-19-02774-t005:** List of thirty metabolites altered between SD and AT.

No	Ret (min)	Metabolites	Unique Mass	Fragment Pattern (*m*/*z*)	ID	Fold Change			
						SSD/SD	LES/SD	MES/SD	HES/SD
Organic acids									
1	5.19	Lactic acid	73	73, 117, 147, 66, 75, 191, 88, 59	STD/MS	1.116	1.050	1.074	1.338 *
2	5.88	Oxalic acid	131	73, 131, 147, 75, 66, 74, 132, 148	STD/MS	1.480 *	0.955	0.955	1.501 *
3	7.41	Phosphoric acid	299	73, 299, 133, 300, 74, 314, 207, 193, 75	STD/MS	0.821	0.705 *	0.784 *	0.632 *
Amino acids									
4	6.78	Valine	218	73, 144, 218, 147, 100, 74, 146, 219, 156	STD/MS	1.185 *	1.103	1.089	1.118
5	7.32	Leucine	158	73, 158, 147, 159, 75, 103, 117, 160	STD/MS	1.554	2.029	1.392	1.610
6	7.54	Isoleucine	158	73, 158, 218, 147, 159, 100, 75, 74	STD/MS	0.975	0.939	0.922	0.957
7	7.67	Glycine	86	73, 174, 86, 147, 100, 175, 59, 248, 176	STD/MS	1.200	1.403	1.350 *	1.365 *
8	8.13	Serine	218	73, 204, 218, 100, 147, 75, 205, 74	STD/MS	1.112	0.947	1.167	1.200
9	8.40	Threonine	219	73, 57, 117, 101, 219, 147, 100, 129	STD/MS	1.029	1.030	1.089	1.121
10	10.41	Phenylalanine	218	73, 218, 192, 100, 147, 75, 74, 219, 193	STD/MS	1.054	1.111	1.032	1.104
11	12.53	Lysine	156	73, 174, 156, 147, 75, 59, 128, 205	STD/MS	1.071 *	1.368 ***	1.282 *	1.539 ***
12	12.65	Tyrosine	218	73, 218, 100, 75, 147, 74, 219, 179, 103	STD/MS	1.057	1.251 *	1.031	1.029
13	14.44	Tryptophan	202	74, 291, 204, 147, 218, 100, 117, 131, 129	STD/MS	1.152	1.267 *	1.031	1.029
Sugars and Sugar alcohols									
14	12.09	Saccharide 1	73	73, 147, 217, 191, 103, 129, 218, 75	MS	1.042	1.001	0.911	0.773
15	12.26	Fructose	103	73, 103, 217, 147, 307, 74, 133, 75	STD/MS	1.303 *	1.695 ***	1.358	1.409 *
16	12.39	Galactose	204	73, 204, 191, 147, 205, 217, 129, 75, 103	STD/MS	0.991	1.139	1.195	1.428 *
17	12.48	Glucose	205	73, 147, 205, 160, 103, 217, 319, 74	STD/MS	0.783	0.651 *	0.762	0.611 *
18	12.60	Glucose	205	73, 147, 103, 205, 160, 129, 217, 157, 319	STD/MS	0.954	0.861 *	0.923	0.787 *
19	12.81	Saccharide 2	217	73, 217, 75, 147, 103, 129, 74, 117, 59	MS	1.357 *	1.809 ***	1.434 *	1.493 *
20	12.94	Saccharide 3	204	73, 204, 147, 191, 217, 75, 205, 74, 129	MS	0.960	1.111	1.102	1.168
21	13.27	Saccharide 4	204	73, 204, 147, 205, 75, 217, 74, 129	MS	0.827 *	0.962 *	1.051	0.790 *
22	13.70	myo-Inositol	217	73, 147, 217, 305, 129, 133, 103	STD/MS	1.568	1.433 *	1.537 *	1.672 *
Fatty acids and Lipids									
23	13.22	Palmitic acid	313	117, 75, 73, 132, 129, 131, 118, 133, 313	STD/MS	1.034	1.246 *	1.267 *	1.143
24	14.40	Stearic acid	341	117, 73, 132, 129, 145, 131, 133, 118, 341	STD/MS	1.175	1.207	1.195	1.014
25	15.40	Oleamide	338	116, 128, 55, 69, 115, 132, 198, 145, 338	STD/MS	0.954	1.071 *	1.002	1.221
26	19.92	Cholesterol	129	81, 55, 107, 57, 105, 91, 121, 119, 93	STD/MS	1.455	1.399	1.252	1.824 *
Non-identification									
27	5.45	N.I.9	72	72, 55, 75, 146, 130, 156	-	0.654	0.556 *	0.594	0.773
28	5.62	N.I.10	116	116, 73, 147, 117, 75, 59, 118, 103	-	1.312	1.154	1.265	1.587 ***
29	6.03	N.I.11	86	73, 147, 133, 59, 86, 100, 72, 89	-	1.011	0.819	1.136	1.111
30	7.64	N.I.12	107	107, 73, 77, 256, 55, 84, 140	-	1.181 *	1.256 *	1.125	1.150
31	7.94	N.I.13	184	184, 73, 134, 77, 86, 59, 100, 285	-	0.864	0.856	0.823	0.771
32	11.44	N.I.14	299	73, 299, 147, 357, 103, 101, 75, 129	-	1.222	1.208	1.161	1.358 *
33	13.51	N.1.15	122	55, 69, 122, 136, 56, 83, 67, 54, 70	-	1.182	1.527 *	1.055	0.998
34	14.98	N.1.16	203	55, 216, 148, 69, 131, 74, 204, 67, 54	-	1.372	2.213 *	1.321	1.750

Metabolites selected by VIP >0.7 and *p* value (<0.05) from OPLS-Da. * *p* < 0.05, *** *p* < 0.001 a significant difference between NS and AT. SD: sleep deficiency, AT: after treatment, SSD: sleep supplement after sleep deficiency, LES: low-intensity aerobic exercise after sleep deficiency, MES: moderate-intensity aerobic exercise after sleep deficiency, HES: high-intensity aerobic exercise after sleep deficiency, OPLS-DA: orthogonal partial least squares discriminant analysis, Ret: retention time, ID: identification, STD: commercial standard compound, MS: to comparison with the mass spectra, *m*/*z*: the selected ion for identification and mass-to-charge ratio.

## Data Availability

The datasets used and analyzed during the current study are available from the corresponding author on reasonable request.

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
