# Peer review of "The Study of Variation of Metabolites by Sleep Deficiency, and Intervention Possibility of Aerobic Exercise"

_ijerph, 2022, doi:10.3390/ijerph19052774_

Round 1
Reviewer 1 Report
Thank you for permitting me to review this manuscript
Major concerns: the methodology should be more simply explained
The introduction is very clear except for primary and secondary objectives which need to be adequately expressed
Methods
Why body temperature of the participants was not considered at all in this study?
Figure 1 please insert the time of exercise
please explain the rationale of the SSD group?
please explain with more basic details in figure 2 and 3
please report the timing of the exercises, was it random or the exercise was performed at the same hour for every participant since this can also affect sleep
results
table 3 and 5 please explain the fragment pattern issue
Table 3 and 4 please explain the p value , which column comparison
Line 391 to 410 please make a better presentation since it is confusing
please insert the matabolomic isssue in the introduction section and delete it in the discussion
conclusion
should be rephrased, with focus only on the conclusion of this study mainly the effect of exercise
It is well demonstrated that SD can attenuate the quality of sleep , induces daytime sleepness and increase fatigue , theses facts are not the specificity of this study
Author Response
Response to Reviewer 1 Comments
Dear Reviewer,
Thank you for the comments on our manuscript entitled " The Study of Variation of Metabolites by Sleep Deficiency, and Intervention Possibility of Aerobic Exercise" (ijerph-1572451). Through the accurate comments made by the reviewer, we better understand the critical issues in this manuscript. We appreciate the reviewer’s comments that helped to revise the manuscript and have revised the manuscript accordingly. Also, our manuscript has been edited by a professional scientific English language editing service. All the changes are highlighted using track changes in the originally uploaded version of the manuscript and this new version is uploaded in the submission system. You can check for line-related information among the response contents below through the “revision version with track change” version file. The detailed responses to the reviewer’s comments are presented as follows:
Point 1: Major concerns - The methodology should be more simply explained.
Response 1: Thank you very much for your suggestion. According to the reviewer's comment, we revised the methodology to be simplified. Also, information of low importance was deleted. (Line 159, 163-165, 181-183, 192-193, 198-199, 200-201, 219-221, 226-229, 233-235)
Point 2: The introduction is very clear except for primary and secondary objectives which need to be adequately expressed.
Response 2: Thank you very much for your suggestion. Based on your comment, we have revised the following to clearly present the primary and secondary objectives. (Line 110~116). The revised contents are shown below:
“Therefore, this study was conducted to identify two purposes. The first purpose was to identify the difference in sleep-related factors (sleep quantity and quality) and metabolites changes between NS (8 hours of sleep per night for 3 consecutive days) and SD (4 hours of sleep per night for 3 consecutive days). The second purpose was to analyze changes in the metabolites according to the difference in aerobic exercise intensity under SD conditions. Through this, we tried to confirm the efficiency and safety of aerobic exercise under SD conditions.”
Point 3: Methods - Why body temperature of the participants was not considered at all in this study?
Response 3: We appreciate the reviewer’s insightful question. It is known that there is a relationship between sleep and body temperature. It would have been a good reference if we had measured body temperature in this study. However, the purpose of this study is not to induce good quality sleep but to examine the changes according to aerobic exercise in the sleep-deprived state. To monitor sleep during the experiment, we tried to obtain many sleep-related variables by wearing a wearable device. In addition, the experiment was conducted under the same conditions (light, noise, room temperature) during the experiment period. With the consent of the reviewers, if the body temperature value is additionally considered in future studies, it will be of great help to improve the quality of the study.
Point 4: Figure 1 please insert the time of exercise.
Response 4: Thank you very much for your suggestion. According to the reviewer's comment, Figure 1 inserted the time of exercise. (Line 184). The revised Figure 1 is shown in the manuscript.
Point 5: Please explain the rationale of the SSD group?
Response 5: Thank you very much for your comment. A nap during the afternoon restores wakefulness and promotes performance and learning. Several studies have shown that naps of less than 30 minutes of duration provide several benefits, whereas longer naps are associated with a loss of productivity and sleep inertia. Also, in this study, the treatment duration between experimental groups was controlled to be the same. Thus, the rationale of the SSD group was added to the manuscript. (Line 260-261). The revised content is shown below:
“Referring to previous studies [30, 31], they slept on a bed in the laboratory to supplement their sleep for 30 minutes after resting.”
30. Dhand, R.; Sohal, H. Good sleep, bad sleep! The role of daytime naps in healthy adults. Curr Opin Pulm Med 2006, 12, 379-382, doi:10.1097/01.mcp.0000245703.92311.d0.
31. Faraut, B.; Andrillon, T.; Vecchierini, M.F.; Leger, D. Napping: A public health issue. From epidemiological to laboratory studies. Sleep Med Rev 2017, 35, 85-100, doi:10.1016/j.smrv.2016.09.002.
Point 6: Please explain with more basic details in figure 2 and 3.
Response 6: We appreciate the reviewer’s comment. We feel sorry for confusing the reviewer. As the reviewer knows, a multivariate analysis must be performed to identify significantly different metabolites among groups. It is meaningful for metabolite analysis when a significant distinction is confirmed in the principal component analysis (PCA) model or orthogonal projections to latent structures discriminant analysis (OPLS-DA). Figure 2 and 3 show the results of multivariate analysis for metabolite analysis. Figure 2(A) shows the results of the principal component analysis (PCA) model. Red-filled triangles mean NS data, and green-filled triangles mean SD data. Figure 2(A) shows that there was no clustering because green triangles and red triangles were mixed. Therefore, for clustering, we performed the orthogonal projections to latent structures discriminant analysis (OPLS-DA) model, which is a higher-dimensional method. Figure 2(b) is the result of OPLS-DA model. As shown in Figure 2(b), it can be confirmed that there is a significant distinction between NS and SD. (see lines 377-381). Figure 4(A) shows the results of the principal component analysis (PCA) model. Green-filled triangles mean SD data, and blue-filled triangles mean data after treatment (AT). As shown in Figure 3(A), there was no clustering because green triangles and blue triangles were mixed. Therefore, for clustering, we performed the orthogonal projections to latent structures discriminant analysis (OPLS-DA) model, which is a higher-dimensional method. Figure 4(b) is the result of OPLS-DA model. As shown in Figure 4(b), it can be confirmed that there is a significant distinction between SD and AT. (see lines 463-468).
Point 7: Please report the timing of the exercises, was it random or the exercise was performed at the same hour for every participant since this can also affect sleep.
Response 7: Thank you very much for your comment. This study was conducted to confirm the effect of aerobic exercise under SD situations. Since the aerobic exercise was performed after 3 days of SD, it did not affect sleep. Additionally, the participants performed the exercise on the date assigned to each subject, however, the exercise start time was the same at 9 am. Based on the reviewer's comments, content related to treatment time was added. (Line 246~247) The added content is shown below:
“Treatment by groups was performed at 9 am on the 4th day after the subject continued SD for 3 days.”
Point 8: Table 3 and 5 please explain the fragment pattern issue.
Response 8: Thank you very much for your comment. The fragment pattern specified Table 3 and 5 means the dissociation of energetically unstable molecular ions formed from passing the molecules in the ionization chamber of a mass spectrometer. The fragments of a molecule produce a unique pattern in the mass spectrum. Fragment pattern can be expressed as a mass to charge ratio (m/z), and through this, the chemical structure, molecular weight, etc. of the material can be identified. Thus, it is possible to confirm what kind of molecular it is through fragment pattern. Some of the related contents are specified in the Method part (Line 326-331), but information has been added to the foot note below the table to help understanding.(Line 387, 478)
Point 9: Table 3 and 4 please explain the p value , which column comparison.
Response 9: We appreciate the reviewer’s comment. We feel sorry for confusing the reviewer. The p-value presented in Table 3 shows a significant difference between NS and SD through an independent t-test. In the fold change, a number greater than 1 means a significantly increased SD than NS, and a number less than 1 means a significantly decreased SD than NS. Meanwhile, the p-value presented in Table 4 shows a significant difference among exercise groups through a one-way ANOVA. Based on the reviewer's comments, content related to p-value was added. (Line 385, 473).
Point 10: Line 391 to 410 please make a better presentation since it is confusing.
Response 10: We appreciate the reviewer’s comment. We feel sorry for confusing the reviewer. According to the reviewer's comment, we revised the contents. (Line 448~460) The revised section is shown below:
“In the SSD, 5 metabolites (oxalic acid, valine, lysine, fructose, and saccharide 2) were significantly higher compared with SD (p<.05). However, 1 metabolite (saccharide 4) was significantly lower compared with SD (p<.05). In the LES, 8 metabolites (lysine, tyrosine, tryptophan, fructose, saccharide 2, myo-Inosistol, palmitic acid, and oleomide) were significantly higher compared with SD (p<.05). However, 3 metabolites (phosphoric acid, glucose, and saccharide 4) were significantly lower compared with SD (p<.05). In the MES, 5 metabolites (glycine, lysine, saccharide 2, myo-Inositol, and palmitic acid) were significantly higher compared with SD (p<.05). However, 1 metabolite (phosphoric acid) was significantly lower compared with SD (p<.05). In the HES, 9 metabolites (lactic acid, oxalic acid, glycine, lysine, fructose, galactose, saccharide 2, myo-Inositol, and cholesterol) were significantly higher compared with SD (p<.05). However, 3 metabolites (phosphoric acid, glucose, and saccharide 4) were significantly lower compared with SD (p<.05).”
Point 11: Please insert the matabolomic isssue in the introduction section and delete it in the discussion.
Response 11: We appreciate the reviewer’s comment. According to the reviewer's comment, we revised the contents. (Line 56~61) The revised section is shown below:
“Metabolomics is a relatively new analytical method within the field of sleep medicine and exercise biochemistry. With the greater ease-of-use and availability of mass-spectrometry equipment, metabolomics has improved access to powerful data acquisition and advanced data processing techniques to detect large numbers of analytes simultaneously in biological samples [12]. Recently, the application of metabolomics in sleep studies is increased due to the advantages of metabolomics.”
Point 12: Should be rephrased, with focus only on the conclusion of this study mainly the effect of exercise.
Response 12: Thank you very much for your comment. We agree with the reviewer’s point. However, as the reviewer knows, our study has two purpses. The first purpose is to identify the difference in sleep-related factors (sleep quantity and quality) and metabolite changes between NS (8 hours of sleep per night for 3 consecutive days) and SD (4 hours of sleep per night for 3 consecutive days). The second purpose is to analyze changes in the metabolites according to the difference in the intensity of aerobic exercise under SD conditions. Therefore, we believe that presenting conclusions related to the first purpose helps to improve the quality of this study. We hope that your opinion has been fully reflected.
Thank you very much for the reviewer’s so many valuable comments. Thanks to the reviewer, the authors have improved their understanding of this research content and the quality of the paper a lot. We hope that the revisions in the manuscript and our accompanying responses will be sufficient to make our manuscript suitable for publication in the International Journal of Environmental Research and Public Health.
Reviewer 2 Report
This work showed an interesting investigation about the difference in sleep-related factors and metabolites between normal sleep and sleep deficiency, and to analyze the variations in metabolites according to the intensity of aerobic exercise in sleep deficiency conditions. The research is well designed and performed, and the manuscript is clearly presented. However, I have a number of further comments to improve the manuscript:
Introduction:
- Add one paragraph for organisation of the paper at the end of introduction section.
- In the introduction section mention that most previous studies have focused on the effects of TSD, and studies on SD are fewer than those on TSD. Therefore, A new section should be created entitled "literature review" or "related works".
Materials and Methods:
- There are many parameters in the proposed method i.e., duration of REM, duration of light sleep, duration of deep sleep, and steps during the day. What's the influence of these parameters?
Results: - Table 4 and related argument: How authors have selected only these variables to use should be commented on in the manuscript.
Discussion:
- Basic sections have been covered, limitations section can be added separately.
Conclusion:
- Basic sections have been covered, future section can be added separately.
Author Response
Response to Reviewer 2 Comments
Dear Reviewer,
Thank you for the comments on our manuscript entitled " The Study of Variation of Metabolites by Sleep Deficiency, and Intervention Possibility of Aerobic Exercise" (ijerph-1572451). Through the accurate comments made by the reviewer, we better understand the critical issues in this manuscript. We appreciate the reviewer’s comments that helped to revise the manuscript and have revised the manuscript accordingly. Also, our manuscript has been edited by a professional scientific English language editing service. All the changes are highlighted using track changes in the originally uploaded version of the manuscript and this new version is uploaded in the submission system. You can check for line-related information among the response contents below through the “revision version with track change” version file. The detailed responses to the reviewer’s comments are presented as follows:
Point 1: Introduction: Add one paragraph for organisation of the paper at the end of introduction section. In the introduction section mention that most previous studies have focused on the effects of TSD, and studies on SD are fewer than those on TSD. Therefore, A new section should be created entitled "literature review" or "related works".
Response 1: We appreciate the reviewer’s suggestion. we agree to add a detailed descriptionof SD-related research to the original text. Based on comments from reviewer, SD-related studies have been added to the introduction. (Lines 74-81). Sections added to the intro:
“Several studies have examined changes in the metabolites of individuals experiencing SD. SD resulted in various changes in circulating metabolites, including increased levels of certain lipids and amino acids, and decreased carbohydrates [18]. In addition, SD is associated with pronounced shifts in the lipid metabolites [19, 20]. To date, however, most previous studies have focused on the effects of TSD, and studies on effects of SD are fewer than those on TSD. In real-life conditions, people experience SD much more than TSD. Thus, it is important to examine the impact of SD in real life.”
18. Bell, L.N.; Kilkus, J.M.; Booth, J.N., 3rd; Bromley, L.E.; Imperial, J.G.; Penev, P.D. Effects of sleep restriction on the human plasma metabolome. Physiol Behav 2013, 122, 25-31, doi:10.1016/j.physbeh.2013.08.007.
19. Depner, C.M.; Cogswell, D.T.; Bisesi, P.J.; Markwald, R.R.; Cruickshank-Quinn, C.; Quinn, K.; Melanson, E.L.; Reisdorph, N.; Wright Jr, K.P. Developing preliminary blood metabolomics-based biomarkers of insufficient sleep in humans. Sleep 2020, 43, zsz321.
20. Weljie, A.M.; Meerlo, P.; Goel, N.; Sengupta, A.; Kayser, M.S.; Abel, T.; Birnbaum, M.J.; Dinges, D.F.; Sehgal, A. Oxalic acid and diacylglycerol 36: 3 are cross-species markers of sleep debt. Proceedings of the National Academy of Sciences 2015, 112, 2569-2574
Point 2: Materials and Methods: There are many parameters in the proposed method i.e., duration of REM, duration of light sleep, duration of deep sleep, and steps during the day. What's the influence of these parameters?
Response 2: Thanks for your insightful question. To increase the reliability of the results of our study, we need to ensure that the participants' sleep is well induced. The sleep-related parameters considered in this study are data measured through wearable devices worn by participants. Therefore, the sleep-related parameters presented in our study are data demonstrating that sleep deprivation was well induced in all sleep stages.
Point 3: Results: Table 4 and related argument: How authors have selected only these variables to use should be commented on in the manuscript.
Response 3: We appreciate the comment. As you know, the variables presented in <Table 4> are commonly used to measure exercise intensity during aerobic exercise. The reason for the measurement in this study was to confirm whether the assigned exercise group performed the exercise well. As shown in <Table 4>, there were significant differences by group in all factors, which means that exercise was performed well for each assigned exercise group. We've added relevant information to the method section (lines 255-257) for readers to understand. Here's the modified how-to section:
"We also measured exercise-related factors (VO2, HR, RER, running speed, calorie consumption, and running distance) to ensure that participants exercised correctly for each group they were assigned to."
Point 4: Discussion: Basic sections have been covered, limitations section can be added separately.
Response 4: Thank you very much for your suggestion. Referring to your comments, the content was separated by creating a limitations section in the discussion. (Line 626~633). The added limitation section is shown below:
“Limitations
There are several limitations to consider when interpreting our results. First, SD stimulation was applied for 3 days in this study. Thus, it is difficult to apply the results to chronic SD or insomnia patients. Second, because the only one-time aerobic exercise was performed in this study, it will be different from the effects of long-term training and anaerobic exercise. Lastly, all participants of exercise groups exercised equally for 30 minutes in this study. Thus, there may be a difference from the effect of long-term exercise.”
Point 5: Conclusion: Basic sections have been covered, future section can be added separately.
Response 5: Thank you very much for your suggestion. With reference to your comment, we added a future section in the conclusion. (Line 643-647). The added future section is shown below:
“Based on the limitations of this study, the following studies should be conducted in future studies. First, future studies in patients with chronic SD and insomnia should be conducted. Second, it should be done to verify the effects of long-term training and anaerobic exercise. Lastly, a study should be conducted to confirm the difference when each group performs with the same amount of exercise.”
Thank you very much for the reviewer’s so many valuable comments. Thanks to the reviewer, the authors have improved their understanding of this research content and the quality of the paper a lot. We hope that the revisions in the manuscript and our accompanying responses will be sufficient to make our manuscript suitable for publication in the International Journal of Environmental Research and Public Health.
Round 2
Reviewer 1 Report
The authors have significantly improved the manuscript